# Convolutional Generation of Textured 3D Meshes

**Dario Pavllo**
Dept. of Computer Science
ETH Zurich

**Graham Spinks**
Dept. of Computer Science
KU Leuven

**Thomas Hofmann**
Dept. of Computer Science
ETH Zurich

**Marie-Francine Moens**
Dept. of Computer Science
KU Leuven

**Aurelien Lucchi**
Dept. of Computer Science
ETH Zurich

## Abstract

While recent generative models for 2D images achieve impressive visual results, they clearly lack the ability to perform 3D reasoning. This heavily restricts the degree of control over generated objects as well as the possible applications of such models. In this work, we bridge this gap by leveraging recent advances in differentiable rendering. We design a framework that can generate *triangle meshes* and associated high-resolution texture maps, using only 2D supervision from single-view natural images. A key contribution of our work is the encoding of the mesh and texture as 2D representations, which are semantically aligned and can be easily modeled by a 2D convolutional GAN. We demonstrate the efficacy of our method on Pascal3D+ Cars and CUB, both in an unconditional setting and in settings where the model is conditioned on class labels, attributes, and text. Finally, we propose an evaluation methodology that assesses the mesh and texture quality separately.

## 1 Introduction

State-of-the-art image synthesis models based on the GAN framework [13] nowadays achieve photo-realistic results thanks to a series of key contributions in this area [19, 34, 35, 65, 25]. A recent trend in this field has been to make generative models more controllable and of better use for downstream applications. This includes works that condition generative models on class labels [35, 65, 4], text [66, 67, 59, 30], input images [70, 22], as well as structured scene layouts such as semantic maps [53, 41, 36], bounding boxes [69, 20, 47], and scene graphs [23]. While these approaches achieve impressive visual results, they are all based on architectures that fundamentally ignore the concept of image formation. Real-world images depict 2D projections of 3D objects, and explicitly considering this aspect would lead to better generative models that can provide disentangled control over shape, color, pose, lighting, and can better handle spatial phenomena such as occlusions. A recent trend to account for such effects has been to disentangle factors of variation during the generation process in the hope of making it more interpretable [63, 44, 25, 26]. These approaches potentially learn a hierarchical decomposition of objects, and in some settings (e.g. faces) they can provide some degree of control over pose. However, the pose disentanglement assumptions made by these approaches have been shown to be unrealistic without some form of supervision [32], and they have not reached the degree of controllability that a native 3D representation would be capable of. More recent efforts have focused on incorporating 3D information into the model architecture, using either rigid transformations in feature space [38] or analysis-by-synthesis [37]. These approaches represent an interesting middle ground between 2D and 3D generators, although their objective remains 2D image synthesis.

In this work, we propose a GAN framework for generating *triangle meshes* and associated textures, using only 2D supervision from single-view natural images. In terms of applications, our approach could greatly facilitate content creation for art, movies, video games, virtual reality, as well as

augment the possible downstream applications of generative models. We leverage recent advances in differentiable rendering [33, 27, 31, 5] to incorporate 3D reasoning into our approach. In particular, we initially adopt a reconstruction framework to estimate meshes through a representation we name *convolutional mesh* which consists of a displacement map that deforms a mesh template in its tangent space. This representation is particularly well-suited for 2D convolutional architectures as both the mesh and its texture share the same topology, and the mesh benefits from the spatial smoothness property of convolutions. We then project natural images onto the UV map (mapping between texture coordinates and mesh vertices) and reduce the problem to a 2D modeling task where the representation is independent of the pose of the object. Finally, we train a 2D convolutional GAN in UV space where inputs to the discriminator are masked in order to deal with occlusions.

Our model generates realistic meshes and can easily scale to high-resolution textures ($512\times512$ and possibly more) owing to the precise semantic alignment between maps in UV space, without requiring progressive growing [25]. Most importantly, since our model is based exclusively on 2D convolutions, we can easily adapt ideas from state-of-the-art GAN methods for 2D images, and showcase our approach under a wide range of settings: conditional generation from class labels, attributes, text (with and without attention), as well as unconditional generation. We evaluate our approach on Pascal3D+ Cars [57] and CUB Birds [52], and propose metrics for evaluating FID scores [19] on meshes and textures separately as well as collectively. In summary, we make the following contributions:

- A novel convolutional mesh representation that is smooth by definition, and alongside the texture, is easy to model using standard 2D convolutional GAN architectures.
- A GAN framework for producing textured 3D meshes from a pose-independent 2D representation. In particular, in a GAN setting, we are the first to demonstrate full generation of textured triangle meshes using 2D supervision from *natural images*, whereas prior attempts have focused on limited settings supervised on synthetic data without a principled texture learning strategy.
- We demonstrate *conditional* generation of 3D meshes from text (with and without an attention mechanism) and show that our model provides disentangled control over shape and appearance.
- We release our code and pretrained models at https://github.com/dariopavllo/convmesh.

## 2 Related work

Deep learning approaches that deal with 3D data typically target either *reconstruction*, where the goal is to predict a 3D mesh from an image, or *generation*, where the goal is to produce meshes from scratch. We review the literature of both tasks as they are relevant to our work.

**3D representations.** Early approaches have focused on reconstructing meshes using 3D supervision. These are typically based on voxel grids [12, 7, 72, 55, 61, 48, 15], point clouds [9], or signed distance functions [40]. However, 3D supervision requires ground-truth 3D meshes, which are usually available in synthetic datasets but not for real-world images. Therefore, a related line of research aims at reconstructing meshes using exclusively 2D supervision from images. Similarly, there has been work on voxel representations [60, 14, 50, 54, 49, 62] as well as on point clouds [21], but these methods require supervision from multiple views which still limits their applicability. More recent approaches lift the requirement of multiple views in order to learn to reconstruct 3D shapes from a single view using a voxel representation [18]. However, these representations tend to be computationally inefficient and do not explicitly support texture maps.

**Differentiable rendering.** Triangle meshes are an established representation in computer graphics, owing to their efficiency as well as flexibility in terms of vertex transformations and texturing. For this reason, they are used in almost every graphics pipeline, ranging from video games to animation. This has motivated a newer line of research where the goal is to predict triangle meshes and texture maps from single images, achieving high-quality visual results [27, 24, 5]. The basic building block of these approaches is a *differentiable renderer* (DR), i.e. a renderer that can compute gradients w.r.t. the scene parameters. While early DRs approximate gradients with respect to mesh vertices [33, 27], newer methods propose fully-differentiable formulations [31, 5]. Our work is also based on this framework, and specifically we adopt DIB-R [5] because it supports UV mapping.

**3D mesh generation.** Analogous to reconstruction methods, 3D object generation has also been demonstrated using voxels [56, 12, 45, 58, 71, 3] and point clouds [1, 11], but again, these approaches require some form of 3D supervision which precludes training from natural images, in addition to the texturing limitations highlighted above. As for triangle meshes, [5] propose a GAN framework

where 2D images are discriminated after differentiable rendering, but they rely on multiple views of synthetic objects and cannot directly learn textures from images. Instead, they supervise the generator on textures predicted by a separate model previously trained for reconstruction. This intermediate step results in a noticeable loss of quality, and is absent in our approach, which can learn from natural images directly. A parallel work to ours [17] also leverages 2D data to generate 3D meshes, but they adopt a VAE framework [29] and only predict face colors instead of UV-mapped textures (i.e. *texture maps*), which limits the visual detail of generated objects. An early work [43] generates untextured meshes in a variational framework using reinforcement learning to estimate gradients. Our work is based on GANs and can explicitly generate high-resolution texture maps which are then mapped to the mesh via UV mapping, enabling an arbitrary level of detail. Unlike [5], we learn textures directly from natural images, and introduce a pose-independent representation that reduces the problem to a 2D modeling task. Finally, we are not aware of any prior work that can generate 3D meshes from text.

## 3 Method

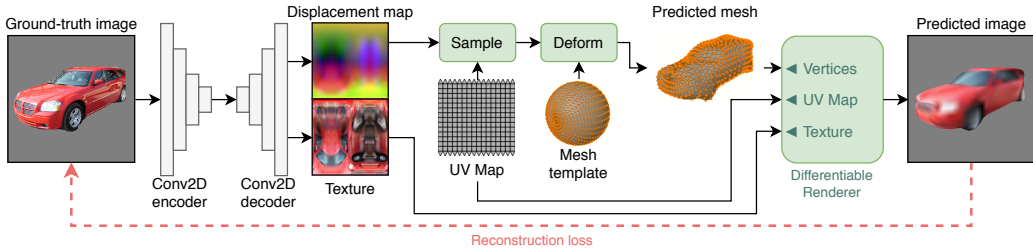

Figure 1: Initial mesh reconstruction using our convolutional mesh representation. This step follows a typical autoencoder setup where the goal is to reconstruct the input image after forcing it through a 3D representation and rendering it. RGB colors in the displacement map correspond to XYZ coordinates.

**Requirements.** Our approach has data requirements similar to recent *reconstruction* methods [24, 5]. We require a dataset of single-view natural images, with annotated segmentation masks and pose, or alternatively, keypoints from which the pose can approximately be estimated. If ground-truth masks are not available (as in the ImageNet subset of Pascal3D+), we obtain them from an off-the-shelf segmentation model (we use Mask R-CNN [16]), whereas the pose is inferred from the keypoints using structure-from-motion, as was done in [24]. Our approach does not require ground-truth 3D meshes (i.e. 3D supervision) or multiple views of the same object.

**Mesh representation.** As mentioned, we focus on triangle meshes due to their wide adoption in computer graphics, their flexibility in terms of vertex transformations, and their support for texture mapping. Following [27, 24, 5], we use a deformable sphere template with a fixed topology and a static UV map which maps vertices to texture coordinates. Previous work has used fully-connected networks to predict vertex positions, which ignores the topology of the mesh and the spatial correlation between neighboring vertices, essentially treating each vertex as independent. This issue is typically mitigated through regularization, e.g. by combining smoothness [27] and Laplacian [46] loss penalties. Instead, we propose to regress the mesh through the same deconvolutional network that we use to regress the texture. The output is therefore a *displacement map* (Fig. 1), which describes how the mesh should be deformed in its tangent space. Importantly, the displacement map and the texture share the same UV map, which ensures that the maps are topologically aligned (e.g. the vertices corresponding to the beak of a bird are co-located with the color of the beak). This detail is crucial for designing a discriminator that can jointly discriminate mesh and texture, that is, not just the mesh and texture separately, but also how well the texture fits the mesh. Furthermore, our mesh representation is smooth by nature since it benefits from the intrinsic spatial correlation of convolutional layers. A second major difference in terms of representation is that our mesh template is a *UV sphere* (2-pole sphere as shown in Fig. 1), whereas prior work has used ico-spheres. While the latter exhibits a more regular mesh, it cannot be UV-mapped without gaps or arbitrary distortions that make the representation space discontinuous. On the other hand, except for the singularities at the two poles, a UV sphere presents a bijective mapping between vertices and texture coordinates, has a well-defined tangent map, and the circular boundary conditions along the $x$ axis can be neatly incorporated in the model architecture using circular convolutions. Denoting the mesh template as $\mathbf{V}$ (an $N \times 3$ matrix with $N$ vertices described by their $xyz$ coordinates, where each vertex is indexed by $i$), the

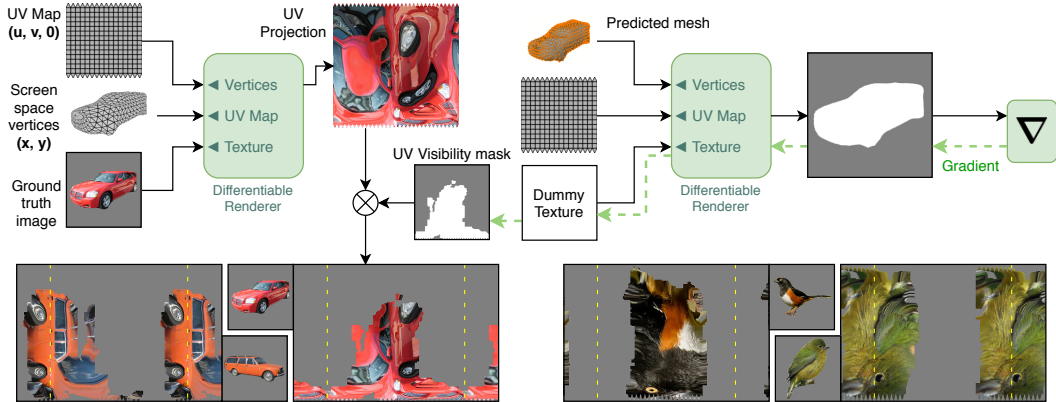

Figure 2: Projection of ground-truth images onto the UV map, producing *pseudo*-ground-truth textures. The bottom row shows additional examples (Pascal3D+ Cars on the left and CUB Birds on the right). The yellow dashed lines represent the boundaries of the textures, which have been extended to highlight the circular boundary conditions along the $x$ axis.

final position of the $i$-th vertex is computed as $\mathbf{v_i} + \mathbf{R_i}\mathbf{\Delta_{v_i}}$ where $\mathbf{\Delta_{v_i}}$ is the output of the model after sampling the displacement map, and $\mathbf{R_i}$ is a precomputed rotation matrix that describes the local normal, tangent, and bitangent of the vertex.

### 3.1 Pose-independent dataset

Our approach initially augments the dataset by estimating a mesh for each training image (Fig. 1). The images are then converted into a pose-independent representation (Fig. 2), which can be finally modeled by a 2D GAN (Fig. 3).

**Mesh estimation.** This is a typical reconstruction task where the goal is to reconstruct the mesh from an input image. Our approach is loosely based on [24], but simplified since we are not interested in performing inference on unseen images. Our formulation can be regarded as a fitting process where we only keep the predicted meshes and discard the model weights/predicted textures. As depicted in Fig. 1, the input image is fed to a convolutional encoder, compressed into a vector representation, and decoded through a convolutional decoder which jointly outputs a texture and a displacement map. The predicted texture is only used to facilitate the learning process and produce more semantically-aligned meshes, and is discarded afterwards. The mesh template is deformed as described by the displacement map, and the final result is rendered using a differentiable renderer. The model is trained to minimize the mean squared error (MSE) between the rendered image and the input image. While this generally leads to blurry textures, it does not represent an issue in our case as these textures are discarded. Since we are not interested in performing inference, we do not predict pose or keypoints, nor do we use texture flows or perceptual losses to improve predicted textures. For the camera model, we adopt a weak-perspective model where the pose of an image is described by a rotation $\mathbf{q} \in \mathbb{R}^4$ (a unit quaternion), a scale $s \in \mathbb{R}$, and a screen-space translation $\mathbf{t} \in \mathbb{R}^2$. For Pascal3D+, we augment the projection model with a perspective correction term $z_0$ (further details in the Appendix A.2). While these are initially estimated using structure-from-motion on keypoints [24], we allow the optimizer to fine-tune $s$, $\mathbf{t}$, and $z_0$ (if used), i.e. we additionally optimize with respect to the dataset parameters[1]. This leads to a better alignment between rendered masks and ground-truth masks, facilitating the next step. As a side note, we mention that inaccurate camera assumptions (e.g. using an orthographic model on photographs that exhibit significant perspective distortion) would most likely not affect the mask alignment or convergence of the model, but might lead to distorted meshes. Nonetheless, our method can work with *any* projection model as long as the camera parameters are known or can be estimated.

**2D discrimination.** The most obvious way to adapt the aforementioned reconstruction framework is to train a GAN where the generator $\mathbf{G}$ produces a 3D mesh and the discriminator $\mathbf{D}$ discriminates its 2D projection after differentiable rendering, as in [5]. However, we found this strategy to lead to training instabilities due to the discrepancies of the representation being used by $\mathbf{G}$ and $\mathbf{D}$ (which are respectively pose-independent and pose-dependent). A further complication we observed is an

aliasing effect in the gradient from the differentiable renderer. Successful 2D GAN models typically use complementary architectures for **G** and **D** (e.g. both convolutional), which motivates our next idea.

**Pose-independent representation.** We instead propose to project ground-truth images onto the UV map of the mesh template, thus reducing the generative model to a 2D GAN that can be trained with existing convolutional techniques. The construction of this representation is depicted in Fig. 2, and can be regarded as a form of *inverse rendering*. We treat our previous mesh estimates as if they were texture coordinates, i.e. $(x, y) \to (u, v)$ ($z$ is dropped), the UV map becomes the mesh to render (a flat surface with $z = 0$), and the texture is the ground-truth image. The result is the projection of the natural image onto the UV map. However, as can be seen in the figure, this process erroneously projects occluded vertices (the back of the car in the example), which should ideally be masked out as visual information associated with them is not available in the 2D image. We therefore mask the projection using a binary *visibility mask*, which describes what parts of the mesh are visible in UV space. The mask is obtained by rendering the mesh using a dummy texture (e.g. all white) and computing its gradient with respect to the texture (we provide implementation details in the Appendix A.2). Only *texels* (pixels of the texture) that contribute to the final image (i.e. visible ones) will have non-zero gradients, therefore we obtain the visibility mask by thresholding these gradients. The final result is a pose-independent dataset of *pseudo*-ground-truth textures (because they are partially occluded). A useful consequence of this representation is that samples become semantically aligned, i.e. the positions of parts such as wheels or eyes are aligned across all images.

## 3.2   GAN framework

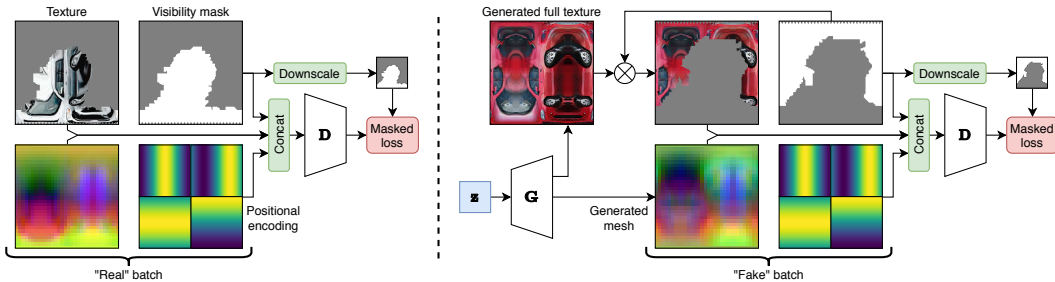

Figure 3: GAN training strategy. **Left:** discrimination of a "real" batch with *pseudo*-ground-truth textures. **Right:** discrimination of a "fake" batch after masking to reflect the "real" batch distribution.

We directly use the estimated displacement maps and the *pseudo*-ground-truth textures to train a convolutional GAN, with the only obstacle that "real" textures are masked, while generated textures should ideally be complete. This can be easily dealt with by masking "fake" images before they reach **D**: as shown in Fig. 3, we multiply the batch of generated textures with a random sample of visibility masks from the training set. This strategy avoids a distribution mismatch between fake and real textures in **D**, while acting as a gradient gate such that only gradients from the visible areas will reach the generator. Being convolutional and agnostic to the visibility mask, **G** will always generate the full texture.

In terms of architecture, the generator is a convolutional model that outputs both mesh (displacement map) and texture. Mesh and texture can have different resolutions – in our experiments we use 32×32 for the mesh and up to 512×512 for the texture. To support this, the generator branches out at some point and outputs mesh and texture through two different heads (this is also done in the *mesh estimation* model). The discriminator adopts a multi-scale architecture [53] (i.e. multiple discriminators trained jointly) and a patch-based loss [22] which is masked using the visibility mask scaled to the same resolution as the last feature map. The smallest discriminator discriminates both the mesh and the texture, which is downscaled to the same resolution as the mesh (32×32). It focuses on global aspects of the texture, while discriminating the mesh and how well it fits the texture. The higher-resolution discriminators are only texture discriminators (one for experiments at 256×256, up to two for experiments at 512×512 in which the intermediate one discriminates at 128×128).

**D** takes as input the displacement map (mesh), the masked texture, the visibility mask, as well as a *soft* positional encoding of the UV map: inspired by attention-based NLP methods that propose a similar idea [51] (this is unrelated to our attention method for text conditioning), we add a sinusoidal encoding to the input that gives convolutions a sense of where they are within the image. For a coordinate space $u, v \in [-1, 1]$, we add four channels $\cos(\pi u)$, $\sin(\pi u)$, $\cos(\pi(v/2 + 0.5))$, $\sin(\pi(v/2 + 0.5))$

such that the encoding smoothly wraps around the $u$ (horizontal) axis and is discontinuous along the $v$ (vertical) axis. Giving an absolute sense of position to the model is important as the semantics of the *texels* depend on their absolute position within the UV map, and we show this quantitatively in the ablation study (sec. 4.3). Finally, the GAN framework allows us to condition the generator on a wide range of inputs: class labels, classes combined with attributes, and text. For the latter, we investigate both an attention mechanism and a method based on a simple fixed-length sentence embedding. We explain how these are implemented in sec. 4.2.

## 4 Experiments

### 4.1 Evaluation and datasets

Perceptual metrics such as the Fréchet Inception Distance (FID) [19] are widely employed for evaluating 2D GANs, as they have been shown to correlate well with human judgment [68]. Although we focus on a different task, the FID still appears as a natural choice as it can easily be adapted to our task. Therefore, we suggest to evaluate FID scores on rendered 2D projections of generated meshes. To this end, we sample random poses (i.e. viewpoints) from the training set as we do not want the evaluation metric to be affected by our choice of poses. Moreover, this strategy allows us to evaluate mesh and texture separately: in addition to the *Full FID*, we report the *Texture FID*, where we use meshes estimated using the differentiable renderer instead of generated ones, and the *Mesh FID*, where we replace generated textures with *pseudo*-ground-truth ones. In the latter, using real poses ensures that we render the visible part of the *pseudo*-ground-truth texture, and occlusions are minimized. While we mostly rely on the *Full FID* to discuss our results, the individual ones represent a useful tool for analyzing how the model responds to variations of the architecture. Generated samples are rendered at $299{\times}299$ (the native resolution of Inception), and ground-truth images are also scaled to this resolution. In the Appendix A.2, we provide some visualizations that give more insight into the conceptual differences between these metrics.

We evaluate our method on two datasets with annotated keypoints, and use the implementation of [24] to estimate the pose from keypoints using structure-from-motion.

**CUB-200-2011 [52]** We use the train/test split of [24], which consists of $\approx$6k training images and $\approx$5.7k test images. Each image has an annotated class label (out of 200 classes) and 10 captions which we use for text conditioning. Using poses and labels (where applicable) from the training set, we evaluate the FID on test images, although we observe that the FID is almost identical between the two sets.

**Pascal3D+ (P3D) [57]** We use the *cars* subset, which is the most abundant class in this dataset. Images are part of a low-resolution set (Pascal set) and a newer, high-resolution set from ImageNet [8]. While we use the same split as [24] to train our mesh estimation model, the GAN is trained only on the ImageNet subset ($\approx 4.7$k usable images) since we noticed that the images in the Pascal set are too small for practical purposes. We infer segmentation masks using Mask R-CNN [16] since they are not available. The test split of [24] does not contain any ImageNet images, therefore we evaluate FID scores on training images [2], motivated by our previous observation on CUB. Finally, to demonstrate conditional generation on this dataset, we collected new annotations for the class (11 shape categories) and color (11 attributes) of each car (details and statistics in the Appendix A.2).

### 4.2 Implementation details

**Mesh estimation.** The model (Fig. 1) is trained for 1000 epochs using Adam [28], with an initial learning rate of $10^{-4}$ halved every 250 epochs. We train with a batch size of 50 on a single Pascal GPU, which requires $\approx$12 hours. We use DIB-R [5] for differentiable rendering due to its support for texture mapping and its relatively low overhead. To stabilize training we adopt a warm-up phase, described in the Appendix A.2. In the same section we also describe how we augment the camera model for Pascal3D+. Finally, the detailed architecture of the network can be found in the Appendix A.1.

**GAN architecture.** Since our method is reduced to a 2D generation task, we adopt recent ideas from the 2D convolutional GAN literature. Our generator follows a ResNet architecture where the latent code **z** (64D, normally distributed) is injected in the input layer as well as after every convolutional layer through *conditional batch normalization*. Following [65, 4], we use spectral normalization [34] in both **G** and **D**, but **D** does not employ further normalization, e.g. we tried instance normalization

but found it detrimental. We adopt a hinge loss objective (patch-based and masked as described in sec. 3), and train for 600 epochs with a constant learning rate of 0.0001 for **G** and 0.0004 for **D** (two time-scale update rule [19]). We update **D** twice per **G** update, and evaluate the model on a running average of **G**'s weights ($\beta = 0.999$) as proposed by [64, 25, 26, 4]. Detailed aspects about the architecture of our GAN can be found in the Appendix A.1. Training the 512×512 models requires ≈ 20 hours on 4 Pascal GPUs, while the 256×256 models require roughly the same time on a single GPU. For all experiments, we use a total batch size of 32 and we employ synchronized batch normalization across multiple GPUs.

**Conditional generation.** In settings conditioned on class labels, we simply concatenate a learnable 64D embedding to **z**, and use *projection discrimination* [35] in the last feature map of **D**. In the P3D experiment with attributes (i.e. colors), we split the embedding into a 32D shape embedding and a 32D color embedding. For text conditioning, we first encode the sentence using the pretrained RNN encoder from [59] (a bidirectional LSTM), and compare *(i)* a simple method where we concatenate the sentence embedding to **z** as before, *(ii)* an attention mechanism operating on all hidden states of the RNN. For the latter we add a single attention layer in **G** right before the mesh/texture branching, operating at 16×16 resolution. Likewise, we modify projection discrimination in **D** to apply attention on the last feature map. Detailed schemes can be found in the Appendix A.1.

**Representation.** Since the UV map of a UV sphere has circular boundary conditions along the horizontal axis, convolutional layers in the discriminator use circular padding horizontally and regular zero-padding vertically. Furthermore, in both the mesh estimation model and the GAN generator, we enforce reflectional symmetry across the $x$ axis as done in [24], which has the dual benefit of improving quality and halving the computational cost to output a mesh/texture. In this case, convolutions use reflection padding horizontally instead of circular padding. Finally, to deal with the singularities of the UV sphere, the vertex displacements of the north and south pole are respectively taken to be the average of the top and bottom rows of the displacement map.

## 4.3 Results

**Quantitative results.** We report our main results in Table 1 (left). For CUB, we compare settings where the model is conditioned on class labels, captions (using the attention model), and no conditioning at all. For P3D, we compare unconditional generation and conditional generation on class

Table 1: **Left:** FID scores grouped by dataset, texture resolution, and conditioning, both in truncated and untruncated settings. Lower is better; bold = best. **Right:** Ablation study on CUB with a 512×512 texture resolution. We report truncated FID scores in the truncated setting.

| Dataset | Tex. res. | Conditioning | $\sigma$ | FID (truncated $\sigma$) | | | FID (untruncated) | | |
|---|---|---|---|---|---|---|---|---|---|
| | | | | Full | Tex. | Mesh | Full | Tex. | Mesh |
| CUB | 512x512 | None | 1 | 41.56 | 45.26 | 18.36 | 56.27 | 50.12 | 25.85 |
| | | Class | 0.25 | 33.63 | 28.68 | 19.49 | **41.33** | **30.60** | 23.28 |
| | | Text | 0.5 | **18.45** | **22.91** | **12.05** | 42.66 | 38.95 | **21.18** |
| | 256x256 | Class | 0.25 | 33.55 | 30.92 | 19.39 | 42.61 | 33.31 | 23.37 |
| P3D | 512x512 | None | 1 | 43.09 | 32.70 | 28.62 | 74.74 | 47.99 | 43.23 |
| | | Class | 0.75 | **27.73** | 22.17 | **23.76** | **49.56** | **29.98** | **34.10** |
| | | Class+Color | 0.5 | 31.30 | **21.70** | 27.75 | 52.55 | 30.29 | 36.32 |
| | 256x256 | Class+Color | 0.5 | 39.09 | 26.52 | 36.73 | 63.63 | 36.56 | 46.37 |

| | FID | Δ |
|---|---|---|
| Baseline (class) | **33.63** | 0 |
| No pos. encoding | 43.71 | +10.08 |
| Same G/D updates | 41.55 | +7.92 |
| InstanceNorm | 36.38 | +2.75 |
| Text with attention | **18.45** | 0 |
| No attention | 22.14 | +3.69 |

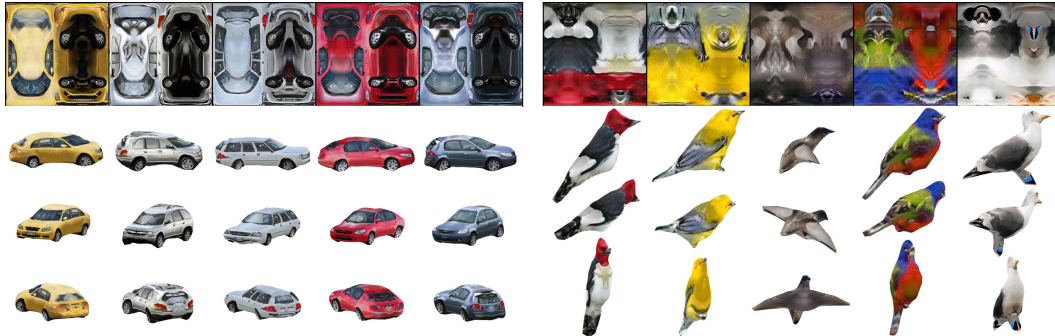

Figure 4: Qualitative results on P3D (left, conditioned on class *and* color) and CUB (right, conditioned on class). Each object is rendered from 3 views, and the top row depicts the unwrapped texture.

labels (i.e. car shapes) as well as classes *plus* colors. We evaluate the FID every 20 epochs and report the model with the best *Full FID* in the table. Since there is no prior work to which we can compare under our setting, we set baselines on these two datasets. As proposed by [4], we found it useful to sample latent codes **z** from a truncated Gaussian (only at inference time), which trades off sample diversity for quality and considerably improves FID scores. For each setting we specify the optimal truncation $\sigma$, but we also report scores in an untruncated setting as these are more directly comparable. As expected, conditional GANs result in better scores than their unconditional counterparts (with the text model being the best), but we generally observe that our approach is stable under all settings.

**Ablation study.** We conduct an ablation study in Table 1 (right). The results in the top section are relative to the $512 \times 512$ CUB model conditioned on classes (truncated FID). Removing the positional encoding from the discriminator leads to a significant FID degradation (+10.08), suggesting that giving convolutions a sense of absolute position in UV space is an important aspect of our approach. Likewise, updating **D** as often as **G** has a significant negative impact (+7.92) compared to two **D** updates per **G** update. Using instance normalization in **D** also leads to a slight degradation (+2.75), but beyond that we observe that, while training appears to converge faster initially, it rapidly becomes unstable. In the bottom section of the table, we compare the text attention model (baseline) to a model where a fixed-length sentence vector is simply concatenated to **z** (as in the other conditional models). The results show that the model effectively exploits the attention mechanism with the added benefit of being more interpretable.

**Qualitative results.** Fig. 4 shows a few generated meshes rendered from multiple views, as well as the corresponding textures. While results on CUB are generally of high visual quality, we observe that the back of the cars in P3D present some artifacts. After further investigation, we found that the dataset is very imbalanced, with only 10–20% of the images depicting the back of the car and the majority depicting the front. Therefore, this issue could in principle be mitigated with more training data. In Fig. 5 we show that the latent space of our models is structured. We interpolate over different factors of variation using spherical interpolation and observe that they are relatively disentangled, enabling isolated control over shape, color, and style in addition to the pose disentanglement guaranteed by the

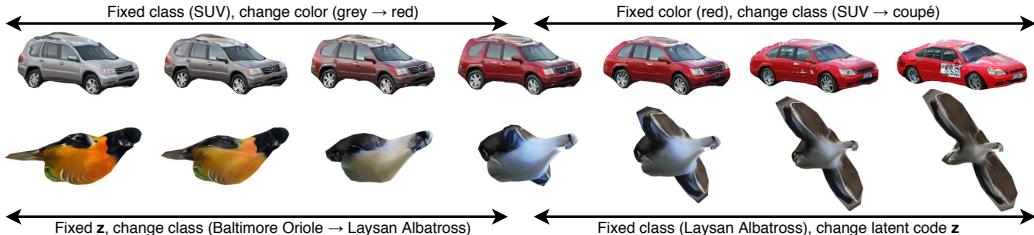

Figure 5: Interpolation over conditioning inputs, which highlights that our model learns a structured latent space where factors of variation of both shape and texture are relatively disentangled.

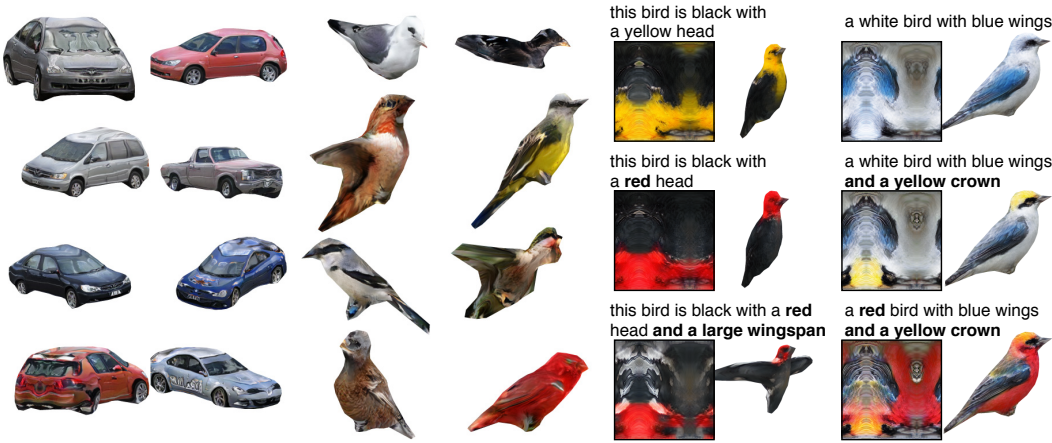

Figure 6: **Left:** generated meshes rendered from random views on P3D and CUB, both conditioned on a random set of classes. **Right:** generation from text on CUB, which allows for fine control over both texture and mesh. We modify captions incrementally, where changes are highlighted in bold.

3D representation itself. Fig. 6 shows results rendered from random viewpoints (the scenario on which we evaluate FID scores) as well as generation conditioned on text, which enables precise control over both shape and appearance. Finally, in the Appendix A.3 we show a wider range of qualitative results.

**2D GAN baseline.** An interesting baseline for generating 3D meshes is to first train a 2D GAN using a state-of-the-art architecture (e.g. StyleGAN [26]), and then run a 3D mesh reconstruction model on top of the generated 2D images. First, we note that such a baseline would not exhibit the properties of a true 3D representation, such as pose disentangled from shape. Additionally, the reconstruction model would have difficulties dealing with occlusions, since it can only reliably infer information visible in the 2D image. To substantiate our observation, we investigate this baseline empirically: we train the 3D reconstruction model of [24] for 1000 epochs on CUB training images with an empty background (our setting). Evaluating this model on *training images* achieves an FID of 85.8 on reconstructions rendered from ground-truth viewpoints, which is already worse than all of our baselines and establishes a lower bound. If we run the model on CUB images produced by StyleGAN [26] and evaluate the FID on renderings from sampled viewpoints, the FID further degrades to 101.9.

**Attention mechanism.** Similar to other attention-based GANs conditioned on text [59], our attention mechanism can be easily visualized. Interestingly, since the attention is applied to our pose-independent representation in UV space and not on flat 2D images, our attention maps can be visualized both in UV space and on 2D renderings, as we show in Fig. 7. Furthermore, our process is more interpretable and semantically meaningful. For instance, prompts that refer to a specific part of the object (e.g. "yellow crown", "red cheeks") activate the same area within the UV map. Most importantly, these correspondences are learned in an unsupervised fashion and are aligned among different images owing to our pose-independent representation.

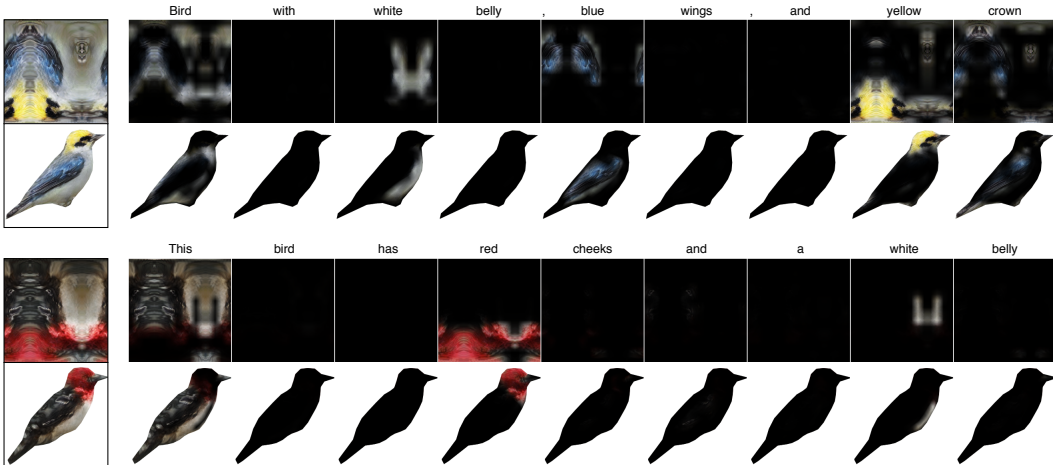

Figure 7: Visualization of the attention mechanism on our CUB model conditioned on text. The attention maps are visualized in UV space (first row) as well as on the rendered mesh (second row). In this particular bidirectional LSTM model, active tokens typically correspond to the adjectives that precede body parts. The first and last tokens are also active because the sentence representation does not comprise explicit sentence delimiters.

## 5   Conclusion

We propose a GAN-based framework for generating 3D meshes. Our approach can generate triangle meshes and corresponding texture maps at high resolution (512×512 and possibly more), requiring only 2D supervision from single-view natural images. We evaluate our method on Pascal3D+ Cars and CUB Birds, and showcase it under a wide range of conditional settings to demonstrate its high level of adaptation. Nonetheless, we have only scratched the surface of what can be done with this framework. Our approach can be enriched by employing different forms of supervision (e.g. semi-supervision by combining 3D supervision from synthetic datasets with 2D supervision from natural images) as well as incorporating more conditional information that would allow the model to disentangle further aspects of variation (e.g. lighting). In the future, we would also like to experiment with larger datasets, and apply the approach to full-scene generation. A viable option is to decompose background and foreground generation as in [42], and use a 3D mesh generator for foreground objects.

## Broader Impact

Our line of research can positively benefit the video game and film industries, both of which impact the life of billions of users. The ability to partially automate the construction of tailored 3D shapes with textures has the potential to reduce costs and timelines by lessening tedious work. The impact on jobs in this area is likely to be minimal as this work is usually performed by specialists whose expertise could be redirected to more creative tasks [2]. Other areas like education and arts could benefit from the ability to bring new concepts to life in an (interactive) 3D environment. Additionally, mesh generation is a hard problem that is likely to be central in many research areas and industry applications going forward.

Adversely, generative models can be used toward fake content creation. The negative societal impact of our method on image generation is likely small as many image modification tools have existed for years [10]. In the longer term, approaches that involve 3D generation might facilitate manipulation of fake video sequences which is harder to achieve with modern software tools. Such applications bring on concerns over exploitation, privacy, political manipulation and the undermining of public institutions. Comparable considerations have already become part of public discourse. A range of approaches have been suggested to tackle such issues ranging from technological, legal, and market solutions. For a more in-depth overview of this discussion we refer to [6].

## Acknowledgments and Disclosure of Funding

This work was partly supported by the Swiss National Science Foundation (SNF), grant #176004, and Research Foundation Flanders (FWO), grant G078618N. Additionally, Graham Spinks and Marie-Francine Moens are partly funded by an ERC Advanced Grant, #788506.

## Footnotes

[1]In an inference model this would be detrimental to generalization, but our goal is mesh fitting.

[2] Given the already small size of the dataset, we decided not to split it further.

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
