[Supplementary Material]

# A Supplementary material

## A.1 Detailed architecture

Figure 8: Generator architecture (left) and mesh estimation model (right). Green blocks comprise learnable parameters, whereas white ones are parameter-free. Dashed lines and blocks in parentheses represent optional connections which depend on the specific setting. We indicate the feature map resolution in a given position next to arrows (e.g. $128{\times}128$). $512 \rightarrow 256$ denotes "512 input channels, 256 output channels". "/2" in convolutional layers denotes "stride 2"; it is one if not indicated.

**Generator.** Fig. 8 (left) shows the detailed architecture of the generator of our GAN. As mentioned in sec. 4.2, the random vector $\mathbf{z}$ is fed to every conditional batch normalization (CBN) layer as well as to the input layer, which matches the strategy adopted by many state-of-the-art GANs for 2D image generation [35, 65, 4]. A CBN layer consists of a parameter-free batch normalization (i.e. without a learned affine transformation) followed by a gain $\boldsymbol{\gamma}$ and bias $\boldsymbol{\beta}$ conditioned upon $\mathbf{z}$ via a learned linear layer. In settings conditioned on class labels, we concatenate a learned embedding $\mathbf{c}$ to $\mathbf{z}$, which is shared among all layers. The network follows a ResNet architecture where feature maps are progressively upsampled using nearest-neighbor interpolation after each residual block `ResBlockG`. This block consists of two convolutional layers wrapped by a skip-connection. If the number of input channels differs from the number of output channels, the skip-connection is learned. To accommodate for the varying output resolutions for mesh and texture, the generator branches out at $32{\times}32$ resolution. While the figure shows the architecture for $512{\times}512$ textures, to generate textures at $256{\times}256$ we simply remove one $256{\rightarrow}256$ `ResBlockG` block. For presentation purposes, we report square resolutions (e.g. $512{\times}512$), but in practice we only need to generate half of the feature map (e.g. $256{\times}512$) since we enforce symmetry across the $x$ axis as mentioned in sec. 4.2. The output textures and displacement maps are then simply padded with their reflection. On the other hand, the discriminator always observes full textures as *pseudo*-ground-truth textures are asymmetric.

**Attention mechanism.** If the model is conditioned on text using an attention mechanism, we add an attention block right before the texture/mesh branch, so that the module influences both mesh and texture. We adopt a dot-product formulation similar to [59], in which the attention weights are computed as $\mathrm{softmax}(\mathbf{QK}^T)$. The queries $\mathbf{Q}$ correspond to the flattened convolutional feature map from the generator, and the keys $\mathbf{K}$ are obtained by passing each RNN hidden state $\mathbf{h}_l$ ($l \in \{1..L\}$, where $L$ is the sentence length) through a learned linear layer. The RNN is the pretrained bidirectional LSTM encoder from [59], and hidden states are 256-dimensional.

**Mesh estimation model.** Fig. 8 (right) shows the detailed architecture of the mesh estimation model that we use for differentiable rendering (the first step of our algorithm as described in Fig. 1). The natural image is concatenated to the segmentation mask (3+1 channels) and fed to a convolutional encoder. The representation is then flattened to a dense representation and passed through a series of linear layers. Finally, it is passed through a ResNet decoder whose architecture resembles that of the GAN generator. Since we are not interested in producing high-quality textures in this step as they are discarded, the texture resolution in this model is only 128×128, which results in faster training.

Figure 9: Multi-scale discriminator architecture for our biggest model (512×512 texture resolution). Only $\mathbf{D_1}$ discriminates the mesh, while $\mathbf{D_2}$ and $\mathbf{D_3}$ are texture discriminators. Dashed lines describe the optional connections for *projection discrimination* [35] in conditional settings, where feature maps are combined *either* with a learned embedding (for settings conditioned on classes or attributes) or with the values of an attention block (for settings conditioned on text), not both.

**Discriminator.** The architecture of our multi-scale discriminator is depicted in Fig. 9. In the most complex setting (used by the CUB model at 512×512), textures are discriminated at three scales: 32×32 ($\mathbf{D_1}$), 128×128 ($\mathbf{D_2}$), and 512×512 ($\mathbf{D_3}$). The smallest discriminator $\mathbf{D_1}$ is also a mesh discriminator. Following the general strategy of patch-based discriminators [22], our discriminators are relatively simple as they only consist of a series of spectrally-normalized convolutional layers. GANs have been shown to produce checkerboard artifacts [39] depending on the choice of kernel sizes and strides in the discriminator. While humans do not perceive these to be particularly severe in images, checkerboard artifacts in the *displacement map* must be avoided as they might lead to noticeable mesh distortions. The generator already uses upsampling instead of transposed convolutions (which mitigates this issue), but we also carefully design the discriminator such that the kernel size of convolutions is divisible by the stride, ensuring that the gradient norms are uniform across pixels (see [39] for further details). To this end, we use 5×5 convolutions in layers with stride 1, and 4×4 convolutions in layers with stride 2. $\mathbf{D_1}$ consists of 4 layers and has a relatively small receptive field. We explored a varying number of layers and different strides, but they always led to worse results. $\mathbf{D_2}$ and $\mathbf{D_3}$ consist of 5 layers and are identical except for the stride of the first layer, which is 1 for $\mathbf{D_2}$ and 2 for $\mathbf{D_3}$. In the experiments with a texture resolution of 256×256, we only use $\mathbf{D_1}$ and $\mathbf{D_2}$, where the latter directly discriminates at 256×256. For Pascal3D+ at 512×512, we found a small empirical advantage in dropping $\mathbf{D_2}$ and doubling the weight of $\mathbf{D_3}$'s loss. Finally, to incorporate conditional information, we use *projection discrimination* [35], in which we compute a pixel-wise dot

product between the last feature map and a learned class embedding (a vector), and add it to the output. If the model is conditioned on text, we replace the class embedding with the output of an attention block. Each discriminator learns its own set of weights for the embeddings or the attention block.

**Model complexity.** Table 2 summarizes the complexity of our models in different settings, expressed as the number of learnable parameters. Since the semantic alignment of our pose-independent representation facilitates the modeling task, our approach can successfully work with relatively small models ($\approx$10M generator parameters). We also found it beneficial to adopt simple discriminators compared to the generator (which is more powerful).

Table 2: Number of learnable parameters for different variants of our model.

| Model | Resolution | G | $D_1$ | $D_2$ | $D_3$ | D Total | RNN |
|---|---|---|---|---|---|---|---|
| Unconditional | 512x512 | 11.58M | 0.68M | 2.77M | 2.77M | 6.22M | - |
| | 256x256 | 10.33M | | | - | 3.45M | - |
| CUB conditional | 512x512 | 13.06M | 0.73M | 2.88M | 2.88M | 6.49M | - |
| | 256x256 | 11.75M | | | - | 3.61M | - |
| CUB text | 512x512 | 11.64M | 0.75M | 2.91M | 2.91M | 6.57M | 2.08M |
| P3D conditional | 512x512 | 13.05M | 0.69M | 2.79M | 2.79M | 6.27M | - |
| | 256x256 | 11.74M | | | - | 3.48M | - |

## A.2 Additional implementation details

**Mesh estimation.** Since our *convolutional mesh* representation already encourages meshes to be smooth, our reconstruction model requires less regularization than similar frameworks based on fully-connected networks. We only found it beneficial to regularize the model with a smoothness loss $\mathcal{L}_{\text{flat}}$ [27] at a very low strength $\alpha = 0.00005$, and no Laplacian regularization [46] (unlike [27, 24, 5] which all use this form of regularization). $\mathcal{L}_{\text{flat}}$ encourages the normals of neighboring faces to have similar directions, and is defined as follows:

$$\mathcal{L}_{\text{flat}} = \alpha \frac{1}{|E|} \sum_{i,j \in E} (1 - \cos \theta_{ij})^2$$

where $E$ is the set of all edges and $\cos \theta_{ij}$ is cosine similarity between the normals of the faces $i$ and $j$. In practice this is implemented by computing the dot product between the two normals.

We additionally observe that the initialization strategy of this model as well as early training iterations have a significant impact on the final result. Bad configurations such as self-intersecting meshes or vertices outside the camera frustum can cause the model to get stuck in bad local minima from which it cannot easily recover. This is especially the case for typical Gaussian initialization schemes in neural networks, which cause the mesh to start in an already self-intersected state for a spherical mesh template with radius 1 (our case). To ensure convergence and generate smooth meshes without self-intersections, we found it helpful to *(i)* zero-initialize the final layer of the mesh branch, which ensures that the first iteration starts with a smooth sphere, and *(ii)* adopt a warm-up phase where $\mathcal{L}_{\text{flat}}$ starts at a moderate strength $\alpha = 0.0005$ and linearly decays for 100 iterations, settling at the low-strength value mentioned above. In the GAN generator we also zero-initialize the final layer of the mesh head, but $\mathcal{L}_{\text{flat}}$ only uses a fixed $\alpha = 0.0001$ and no warm-up.

In sec. 3, we mention that our camera projection model is a weak-perspective model. This model is a good approximation for photographs shot with high levels of zoom or that depict small objects, which is the case for birds (CUB dataset). However, we observed that the weak-perspective assumption is not a good fit for Pascal3D+, since most images are shot from a close range and present a significant degree of perspective distortion due to cars having elongated shapes. Therefore, for Pascal3D+ we augment the camera model with a learnable perspective correction term $z_0$, without however advancing to a full perspective model as we do not have enough information. $z_0$ is a scalar that describes the distance from the camera to the center of the object, and assumes that the object is centered. The $x, y$ coordinates of each vertex in camera space are then multiplied with a factor $(z_0 + z/2)/(z_0 - z/2)$, where $z$ is the depth of the vertex. Note that, as $z_0$ approaches infinity, the factor approaches 1 and the camera model reverts back to a weak-perspective model. This term is learned for every image in the dataset and is parameterized as $z_0 = 1 + e^w$ ($w$ is a learnable parameter), which ensures *(i)* positivity, and *(ii)* that the transformed vertices lie inside the camera frame. While this aspect is not central to our approach, we found it helpful as it can slightly improve qualitative results even with approximate estimates.

Figure 10: Examples of images on which we compute FID scores. Images are rendered from the viewpoint corresponding to *Real image* (a randomly-selected image from the training set). In the *Mesh FID* scenario, we render the generated mesh using the *pseudo*-ground-truth texture from the real image. In the *Texture FID*, the "real" mesh is textured using the generated texture. In the *Full FID* and *Mesh FID* of the top-left van we can observe that the silhouette of the mesh looks fine but straight lines and stripes present a "wobbling" effect caused by the underlying mesh, while in the *Texture FID* (which does not use generated meshes) the lines appear more straight.

**Construction of the pose-independent representation.** In sec. 3 we mention that we use the gradient from the differentiable renderer to produce the UV visibility mask which is used for masking projected textures. In practice, deep learning frameworks do not compute full Jacobians but only gradients of scalars (i.e. Jacobian-vector multiplication). However, the Jacobian of the rendering operation w.r.t. the texture has a structure such that it is zero for all texels that are not visible in the rendered image (i.e. are occluded) and non-zero elsewhere[3]. Based on this observation, it suffices to compute the average or sum of rendered pixels to reduce the image to a scalar which can then be differentiated with respect to a dummy texture. The same result can also be achieved by computing a Jacobian-vector multiplication with a vector of ones, which is what we do in our implementation.

**FID evaluation.** To give more context to sec. 4.1, where we introduce our evaluation methodology, in Fig. 10 we show some actual examples of rendered images on which we compute FID scores. The *Full FID* (our main metric) is computed on generated meshes coupled with generated textures, and evaluates the generation quality as a whole. However, it is also interesting to propose variations of this metric that can evaluate mesh and texture quality separately. Therefore, in the *Mesh FID* we use the *pseudo*-ground-truth texture from the image corresponding to the random viewpoint we choose for rendering, which makes the evaluation independent of generated textures. Likewise, in the *Texture FID* we use meshes estimated using the differentiable renderer instead of the ones generated by our GAN. In all experiments, we generate as many images as there are in the set we compare to, since the FID is sensitive to the number of generated images. Finally, to evaluate text conditioning on CUB, we sample a random caption (out of 10 captions) for each image we generate.

**Pascal3D+ annotations.** To demonstrate conditional generation on P3D, we collected shape and color annotations for the ImageNet subset of this dataset (i.e. the one we use to train our GAN). Although ImageNet images are already identified by their *synsets*, we found these to be unreliable and opted instead for collecting our own annotations. The set of labels and corresponding frequencies are summarized in Table 3. For consistency, all labels were collected by one annotator. Some categories (e.g. *F1*, *convertible*, and *oldtimer*) comprise a very low number of samples, which leads to unsatisfactory results on these classes in conditional settings. Nonetheless, this issue can be mitigated by collecting more data. Finally, although the ImageNet subset consists of ≈5.5k images, only 4.7k are usable as some are filtered out by the structure-from-motion routine of [24] due to unreliable pose estimates.

Table 3: Relevant statistics for the P3D annotations we collected.

| Class | Sedan | Hatchback | SUV | Station wagon | Van | Pickup | Coupé | City | F1 | Convertible | Oldtimer | Total |
|---|---|---|---|---|---|---|---|---|---|---|---|---|
| # images | 1137 | 851 | 814 | 691 | 674 | 649 | 295 | 193 | 119 | 39 | 13 | 5475 |

| Color | Gray | Black | Red | White | Blue | Green | Yellow | Orange | Brown | Purple | Pink | Total |
|---|---|---|---|---|---|---|---|---|---|---|---|---|
| # images | 1534 | 863 | 833 | 832 | 697 | 252 | 231 | 126 | 52 | 30 | 25 | 5475 |

Figure 11: Generation on P3D with one varying factor at a time (color and shape) and a fixed random vector **z**. As can be seen, representations are relatively disentangled. In the bottom row, the class *coupé* is often associated with race cars, which causes stickers to appear on the body.

## A.3 Additional results

**Disentanglement.** Compared to a generative model for 2D images, a 3D generative model naturally disentangles pose and appearance. Furthermore, the use of *triangle meshes* with UV-mapped textures ensures that shape is disentangled from color, essentially allowing for texture transfer between meshes (we use this property for our *Texture FID* and *Mesh FID* evaluation, as shown in Fig. 10). Another interesting observation is that conditional models enable further disentanglement of aspects of variation. For instance, on P3D we can control shape and color separately as shown in Fig. 11, and the latent space is structured enough to allow for interpolation of these aspects (Fig. 5). In this setting, the random vector **z** can be used to control the style of the object.

**Additional qualitative results.** In Fig. 12, we show additional qualitative results grouped by type of conditioning. Our approach successfully generates meshes in both conditional and unconditional settings. In the figure, we additionally show untextured (i.e. *wireframe*) meshes, which highlights the smoothness of our convolutional mesh representation.

**Demo video.** The supplementary material includes a video where we show more results, including latent space interpolation, disentangled generation, generation from text, and visualization of the attention mechanism on models conditioned on text.

P3D unconditional

CUB unconditional

P3D conditioned on class

CUB conditioned on class

| Hatchback | SUV | Pickup | Hatchback |

| Eastern Towhee | Loggerhead Shrike | American Goldfinch | Fox Sparrow |

P3D conditioned on class+color

CUB conditioned on text

| SUV+Red | Hatchback+Blue | Sedan+Black | Hatchback+Gray |

| This bird is white and black with a solid white throat, breast, and abdomen | This tiny bird has a blue head, blueish gray wings, and a bright yellow breast and belly | Bird has orange beak, white belly, the rest of the bird is black | This vibrant bird features red on its belly, blue on its head, and yellow on its back |

Figure 12: Qualitative results for all settings, on both P3D (left) and CUB (right). First row = texture; second row = wireframe mesh; third and fourth rows = textured object from two random views.

## Footnotes

[3]This property holds for DIB-R [5] but may not hold for all differentiable renderers.