[Reviews · NeurIPS 2020]

Review 1

Summary and Contributions: This work proposes a generative model for textured 3D meshes based on the popular GAN framework. At its core is the idea of parameterising both texture and surface as 2D representations, so that to handle them with 2D convolutional GANs. Interestingly, supervision is obtained by leveraging recent advances in single view reconstruction through differentiable rendering.

Strengths: In my opinion, the most interesting contribution of the paper is in the proposed pipeline to generate data for training the generative adversarial network: - I find the idea of using SVR reconstruction pipeline to generate "ground truth" geometry in an unsupervised fashion to be clever, and interesting for the community at large - Similarly, the idea of generating pseudo-ground truth textures by projecting the ground truth image onto the UV map using a simple orthographic projection is interesting and relevant for future work towards generation of textured 3D surfaces

Weaknesses: Qualitative results, especially for the car category, show that there is still some work to do to achieve the generation of high quality textured 3D surfaces: - The first clear problem is in the ground truth geometry: the proposed representation can only capture fixed topology (i.e. the one of a sphere) and struggles at modelling high frequency details. It would interesting to see which improvement could be achieved by using a stronger prior on geometry (as suggested by the authors in the conclusion, enforcing a stronger prior on geometry through synthetic datasets would be helpful - e.g. pre-training the decoder on shapenet cars) - I wonder how the simple orthographic projection models impacts the alignment of geometry and texture, as the applied transformation differs significantly from the "real" one. Could the author give insight on this?

Correctness: Yes, the pipeline is correct, novel, and interesting.

Clarity: Yes, the paper is well written and easy to follow. The supplemental video further demonstrates the result achieved by the proposed pipeline.

Relation to Prior Work: The related work section analyses related work in detail.

Reproducibility: Yes

Additional Feedback: Thanks to the authors for their response, which addresses my concerns.


Review 2

Summary and Contributions: Recently, inferring 3D geometry of objects has drawn great attention while generating textured meshes for objects still lacks research studies. This work is a pioneer attempt. To simultaneously generate the mesh vertices and texture maps, the proposed method utilizes a unified uv-parameterization to encode the appearance for texture and displacement map for geometric information. The displacement map is defined over a given template sphere mesh. The results look appealing.

Strengths: - This is really a timly work which is a pioneer attempt to generate textured meshes from a single-view image. - The proposed network architecture is novel to me. It also makes a lot of senses. Using uv-map for textured mesh generation is a smart strategy. The results also demonstrate the effectiveness. - The obtained results seem appealing to me which are obviously better than previous works I have seen.

Weaknesses: - My major concern is the lack of comparisons against other methods. In my knowledge, there are several baselines: 1) generating 3D voxels with color channels using 3D CNNs; 2) in pix2mesh framework, adding a separate branch to infer the color for each vertex. There are no comparisons against such baselines, thus it is not clear how does the proposed method perform. - There is an obvious limitation that the proposed method cannot deal with the shapes with complex topologies. The examples shown in the paper are all of simple structures. I hope to see whether the method can work for complicated objects like chairs. I guess this would be very challenging. Although there are these weaknesses, the proposed approach well advances such a emerging and timely research topic.

Correctness: Yes, they are correct to me.

Clarity: Yes.

Relation to Prior Work: Yes.

Reproducibility: Yes

Additional Feedback: Post-rebuttal: I am also satisfied with the rebuttal, it addresses most of my concerns. Even the current version has some limitations it cannot handle complex shapes), it can be a good starting to trigger future research in this direction.


Review 3

Summary and Contributions: The paper presents a framework to generate triangle meshes and texture maps which can be learned using single-view natural images. Given a set of 2D images, the images are augmented with mesh estimates and subsequently converted to pose-independent 2D representation via UV-projection. Finally, a 2D GAN is learned to generate 3D meshes where, given a random vector, the generator generates displacement maps and textures, and discriminator discriminates between real/fake displacement maps and textures. Experiments are performed on CUB and Pascal-3D+ datasets.

Strengths: - The generation of 3D meshes is a challenging problem and of great interest to the community. - The proposed approach is novel and makes sense. - The proposed convolutional mesh representation is novel and intuitive. - I like the rationales behind choosing UV-sphere for mesh representation. They are well motivated and intuitive. - I like the usage of positional encoding which yields higher FID scores as shown in the ablations studies. - The proposed approach can generate meshes conditioned on class labels, attributes, and labels. - The qualitative results provided in supp. material look very good. - Sufficient experiments are performed to satisfy design choices. In summary, it is a very good paper and will definitely justify acceptance to NeurIPS.

Weaknesses: - I cannot find any signficant limitation of the paper.

Correctness: Yes

Clarity: - The paper is well written and easy to read. - Sufficient implementation details are provided. - The authors will release the source code which solves the reproducibility problem.

Relation to Prior Work: Yes. Some remarks: Line 26-28 "Provide a limited degree of control over pose": The statement is not quite true and the following papers should be cited: HoloGAN: Unsupervised learning of 3D representations from natural images, ICCV'19 Self-Supervised Viewpoint Learning from Image Collections, CVPR'20 HoloGAN shows that they can obtain control over poses in many diverse scenarios.

Reproducibility: Yes

Additional Feedback: Typos: Line-134: consists in -> consists of ## Post rebuttal. I didn't have any critical remarks in my initial reviews. Other reviewers also agree that it's a good paper and should be accepted to NeurIPs. I keeping my original rating.


Review 4

Summary and Contributions: This paper proposes a generative adversarial network for generating 3D meshes and textures. It first trains a network to predict meshes and textures from a single image, and then it uses the meshes and images to generate ground truth training data to train a GAN. The experiment results show that the proposed method can generate new textured 3D meshes from sampled latent code and text descriptions.

Strengths: 1. The method is able to generate new 3D meshes with textures from sampled latent codes and text. 2. The method uses 2D displacement maps to represent 3D meshes, which enables it to use 2D convolutions to generate 3D contents.

Weaknesses: 1. While I find the combination of single-image 3D reconstruction and GAN interesting, I am concerned about the technical contribution of the paper. It seems that each component is similar to previous works. The single-image 3D reconstruction network is almost identical to [24], and the GAN network also are standard. It feels like the contribution of the paper is just a combination of these two tasks. 2. Another solution to the proposed task here is that first training a 2D GAN to generate new 2D images of specific category, and then directly run the single-image reconstruction network such as [24] to generate textured mesh from the input image. The paper should include a comparison to this baseline. My sense is that currently GAN can generate very high-quality 2D images from sampled latent codes and text. It should be easy to directly generate resonable textured meshes from the high-quality 2D images. It is not clear to me why the proposed framework would outperform this baseline, considering that the performance of the proposed method is also bounded by the performance of single-image 3D reconstruction network. In addition, this alternative solution would be more flexible than the proposed method, since you can use arbitrary GAN network to generate 2D images without re-training the reconstruction network. 3. Why using the sinusoidal encoding in the network? How does it compare to directly using the (u, v) coordinates? Overall, I like the results of the paper. However, I am not fully convinced about the choice of the framework, particularly for the questions discussed in point 2. The technical contributions of the proposed method is also not significant to me.

Correctness: The paper is technically sound.

Clarity: The paper is well written.

Relation to Prior Work: The reference is good.

Reproducibility: Yes

Additional Feedback: Post-rebuttal: The rebuttal addresses my concerns, and I increase my score and agree for accepting the paper.

[Author Response · NeurIPS 2020]

We thank the reviewers for their time and the overall positive feedback. We address each reviewer separately:

• **R1:** "the proposed representation can only capture fixed topology [sphere] and struggles at [..] high frequency details.

[..] would be interesting to see which improvement could be achieved by using a stronger prior on geometry (as suggested

by the authors [..], enforcing a stronger prior on geometry through synthetic datasets would be helpful [..])" This is a valid

point. Although our method can work with other topologies as long as they are fixed, dynamically varying the topology is

challenging. Nonetheless, we believe our approach represents a first step in the right direction and the framework is flex-

ible enough to support extensions (e.g. semi-supervision), as both R1 and we observe. "how the simple orthographic pro-

jection models impacts the alignment of geometry and texture " We partly discuss this in the appendix A.2, 3rd paragraph.

On Pascal3D+ cars, we add a learned perspective correction term to address foreshortening (unnecessary on CUB as ob-

jects are distant). Geometry/texture are always aligned since the initial reconstruction model optimizes for this task, but

inaccurate camera assumptions might lead to distorted meshes. We will discuss this more in depth in the revision. We also

point out that our method can work with *any* projection model if the camera parameters are known or can be estimated.

• **R2:** "comparisons [..] 1) generating 3D voxels with color channels using 3D CNNs; 2) [..] branch to infer the color

for each vertex." We would first like to emphasize that voxels do not support texture mapping, which is a key aspect of

our work, and which requires a triangle mesh setting. Textured triangle meshes are widespread in graphics (e.g. movies,

games) and are crucial for achieving high detail, since simple meshes can be coupled with high-res textures; Additionally,

*(i)* textures are more general than vertex colors (in our method, a $32\times32$ texture is equivalent to vertex colors); *(ii)* while

it is easy to implement the proposed baselines using 3D supervision from synthetic data, it seems unclear how to translate

them to a GAN framework using solely 2D supervision (our setting). Some voxel approaches use 2D supervision, but

they only focus on geometry (no texture/colors) or reconstruction (not generation). *(iii)* In voxels, shape and color are not

disentangled, which limits their generalization and flexibility in ways that are not captured by quantitative metrics (e.g.

texture transfer is not possible). "shapes with complex topologies [..] complicated objects like chairs." We agree this is

challenging. One potential direction is to use semi-supervision, as we observe in the conclusion (also mentioned by **R1**).

Nonetheless, we observe that our model can at least handle some complex features (e.g. beaks, tails and wings in CUB).

• **R3** does not observe any major issue ("I cannot find any significant limitation of the paper") and appreciates our design

choices ("I like the rationales behind choosing UV-sphere for mesh representation [..] well motivated and intuitive", "I

like the usage of positional encoding which yields higher FID scores as shown in the ablations", "it is a very good paper

and will definitely justify acceptance to NeurIPS"). "Some remarks [Prior work]" We thank R3 for those references and

we will discuss them in the revision. "Typos" Thank you, we will take care of that in the revision.

• **R4:** "While I find the combination of single-image 3D reconstruction and GAN interesting, I am concerned about the

technical contribution" We thank R4 for their constructive critical feedback. We think however that the raised concerns

are not directed at our core contributions, which include *(i)* a novel convolutional mesh representation whose strengths

include smoothness, semantic alignment, and the ability to be modeled using existing 2D convolutional GAN strategies.

Thanks to the latter, our method can be easily adopted by the community and can benefit from future advances in the field.

*(ii)* A way to learn textures directly from image pixels using an inverse rendering approach and masking. *(iii)* A *full* frame-

work for textured mesh generation with supervision from natural images, whereas prior work has focused on more limited

settings. Our contributions are also acknowledged by **R1** ("correct, novel, and interesting", "interesting for the commu-

nity at large", "relevant for future work"), **R2** ("work is a pioneer attempt", "network architecture is novel", "results [..]

are obviously better than previous works", "approach well advances such a emerging and timely research topic"), and **R3**

("approach is novel and makes sense", "great interest to the community", "convolutional mesh representation is novel

and intuitive"). We would be grateful if R4 could reconsider their position in light of our response. "**[1/4]** Another solu-

tion to the proposed task [..] first training a 2D GAN to generate new 2D images [..] then directly run the single-image

reconstruction network such as [24]" This is an interesting baseline, although we find it has some limitations compared

to our native 3D representation (see [2/4]). First, we investigated it empirically: *(i)* running [24] on CUB *training images*

achieves a FID of 85.8 on reconstructions, which is already worse than our scores and sets a lower bound; *(ii)* when run-

ning [24] on CUB images produced by StyleGAN, the FID further degrades to 101.9. The setting is the same as ours (no

background). "**[2/4]** why the proposed framework would outperform this baseline" The baseline would not have the prop-

erties of a true 3D representation, e.g. pose disentangled from shape. The reconstruction model would perform subopti-

mally on occlusions (the model can only reliably infer information visible in the 2D image), and interpolating in the latent

space of the 2D GAN would affect the 3D result unevenly. Contrarily, with our pose-independent UV representation, tex-

tures/vertices are semantically aligned across different images (i.e. parts such as wheels always show up in the same posi-

tion), which greatly facilitates learning. "**[3/4]** the performance of the proposed method is also bounded by the [..] single-

image 3D reconstruction network." Our reconstruction network is only used for the geometry, but textures are learned

directly from images in a pure GAN setting (while reconstruction methods often use a VGG loss to avoid blurry textures).

"**[4/4]** this alternative would be more flexible [..] since you can use arbitrary GAN to generate 2D images without

re-training the reconstruction network." For best results with the proposed alternative, the reconstruction model would

still need to be retrained on the same dataset. "Why [..] sinusoidal encoding [..]? [..] directly using (u, v) coordinates"

Our UV map has circular boundary conditions, so concatenating $\sin,\cos(\pi(u,v))$ is more principled as it smoothly wraps

around the edges. This is however a minor technical point and concatenating plain (u,v) coordinates might also work.

[Meta-Review · NeurIPS 2020]

Four knowledgeable referees support accept and I accept. We encourage and expect the authors to incorporate the reviewers' suggestions for improving the paper.